# Application of Ionic Liquids for Batteries and Supercapacitors

**DOI:** 10.3390/ma14112942

**Published:** 2021-05-29

**Authors:** Apurba Ray, Bilge Saruhan

**Affiliations:** German Aerospace Center (DLR), Department of High-Temperature and Functional Coatings, Institute of Materials Research, 51147 Cologne, Germany; apurba.ray@dlr.de

**Keywords:** ionic liquids, li-ion battery, supercapacitor

## Abstract

Nowadays, the rapid development and demand of high-performance, lightweight, low cost, portable/wearable electronic devices in electrical vehicles, aerospace, medical systems, etc., strongly motivates researchers towards advanced electrochemical energy storage (EES) devices and technologies. The electrolyte is also one of the most significant components of EES devices, such as batteries and supercapacitors. In addition to rapid ion transport and the stable electrochemical performance of electrolytes, great efforts are required to overcome safety issues due to flammability, leakage and thermal instability. A lot of research has already been completed on solid polymer electrolytes, but they are still lagging for practical application. Over the past few decades, ionic liquids (ILs) as electrolytes have been of considerable interest in Li-ion batteries and supercapacitor applications and could be an important way to make breakthroughs for the next-generation EES systems. The high ionic conductivity, low melting point (lower than 100 °C), wide electrochemical potential window (up to 5–6 V vs. Li^+^/Li), good thermal stability, non-flammability, low volatility due to cation–anion combinations and the promising self-healing ability of ILs make them superior as “green” solvents for industrial EES applications. In this short review, we try to provide an overview of the recent research on ILs electrolytes, their advantages and challenges for next-generation Li-ion battery and supercapacitor applications.

## 1. Introduction

With the remarkable development of low-cost, lightweight, portable electronic devices, electrical vehicles, aerospace and medical systems in our daily lives, the demand for a sustainable energy supply is becoming one of the greatest challenges for the whole world [1,2,3,4]. In the search for advanced sustainable energy sources, the urgent development of long-term high-efficient high energy-power resources is required. Electrochemical energy storage (EES) systems such as batteries and supercapacitors are promising for reaching the required demands due to their fast charge–discharge rates, high power ability, long cycle life for supercapacitors and high-rate capability and high energy density for batteries [5,6]. Among different types of rechargeable batteries, lithium-ion (Li-ion) batteries have been considered as one of the most promising EES sources on the market since 1991. The charge storage mechanism of Li-ion batteries is mainly based on intercalation/deintercalation of Li-ion between cathode and anode electrodes separated by an electrolyte (Figure 1a). Li-ions move from cathode to anode via electrolyte to store energy during charging and move back to the cathode to release energy during discharging of the Li-ion battery. The movement of Li-ions is facilitated by an electrolyte, which usually contains a highly flammable and volatile solvent. Therefore, safety is a great issue for Li-ion batteries due to thermal instability, leakage possibility, internal short circuit, and the flammability of electrolytes [7,8]. On the other hand, supercapacitors (SCs) draw a lot of interest due to their high-power densities, rapid charge–discharge capability, long cycle lives and, most importantly, safety, which make their use in promising high performance energy storage devices [9,10,11,12]. Depending on the charge storage mechanism, SCs can be categorized into three types, i.e., electrical double-layer capacitors (EDLCs), pseudocapacitors and hybrid supercapacitors (Figure 1b). EDLCs can store energy by a non-Faradic process, i.e., charges are accumulated at the electrode/electrolyte interface in the form of an electric double layer. Carbon-based materials such as carbon nanotube (CNT), activated carbon (AC), graphene, carbon aerogel, etc., are used as EDLC electrode materials due to their high specific surface area, high electrical conductivity and porous structure [10,13]. Still, EDLCs are unable to exhibit high specific capacitance (C_sp_) and high energy density according to the market requirements. On the contrary, different types of binary, ternary transition metal oxides (TMOs) (RuO_2_, NiO, Fe_2_O_3_, MnO_2_, CoO_x_, NiMnO_2_, Co_3_O_4_, MnFe_2_O_4_, etc.), conducting polymers (CPs) (polyaniline (PANI), polypyrrole (PPy) and poly(3,4-ethylenedioxythiophene) (PEDOT), etc.) and their composites (NiO-CNT, V_2_O_5_-PANI, PANI-CNT, NiMn_2_O_4_-Graphene, etc.) are usually used as pseudocapacitive electrode materials [2,14,15,16,17]. Pseudocapacitor electrodes can store energy via reversible faradic redox reactions or ion intercalation into layered structures of electrodes at the electrode/electrolyte interface, and they can reach high specific capacitance and energy density compared to EDLCs [18,19,20]. Hybrid supercapacitors can store charge in the combination of EDL and redox reactions. SCs, however, can deliver high specific capacitance, high power and energy density but poor rate capability, lower electrochemically stable potential windows, lower thermal stability as well as poor charge–discharge stability due to low electrical conductivity and degradation of electrode materials during long cycle life, hinder their market application as energy storage systems (ESSs) [21,22].

Thus, for high-performance EES devices that can operate at higher potential windows and a wide range of operating temperatures, electrolytes play a significant role. Electrolytes can considerably influence the device performance of batteries and supercapacitors. A wide potential window can enhance the energy density of a device and high ionic conductivity, low electrolyte resistivity, low viscosity, etc. It can also improve the power density of a device. The choice of suitable electrolytes for EESs is one of the key challenges for researchers and industries. Electrolytes are commonly divided into three categories such as solid electrolytes, polymer electrolytes and liquid electrolytes. Solid electrolytes (SEs) exhibit high intrinsic oxidation stabilities and comparatively satisfactory electrochemical stabilities, which make them a promising candidate for EES devices, but their brittle nature often restricts them from integrating with EES devices [25,26]. The flexibility and versatility in the shape of polymer electrolytes (PEs) attract a massive interest in various EES devices. Polyethylene oxide (PEO) with Li-salts is mostly used as polymer electrolytes in LIBs. However, the low ionic conductivity at moderate temperatures for PEs binds their practical applications [27,28]. Conversely, the traditional liquid electrolytes possess high ionic conductivity and good compatibility with electrodes, but flammability, fire hazard and catastrophic explosion are concerns for the safety features of the electrolytes for EES devices [29,30]. Usually, there are three types of liquid electrolytes such as aqueous electrolytes, organic electrolytes and ionic electrolytes used for EES devices. Though aqueous electrolytes (such as KCl, KOH, HCl, H_2_SO_4_, NaCl, etc.) provide higher ionic concentration and conductivity, they are limited to a lower potential window due to H_2_O decomposition potential (1.2 V), this significantly diminishes the energy density as well as the electrochemical stability of the device. Most of the acetonitrile (ACN), propylene carbonate (PC), etc., solvent-based organic electrolytes are limited to an even lower potential of up to 3.5 V. High cost, toxicity and safety concerns also restrict the commercial application of organic electrolytes. To overcome such drawbacks, a lot of research interest in the last few decades has been devoted to ionic liquids for next-generation EES device application due to low volatility, non-toxicity, non-flammability, good electrochemical stability (up to 6.0 V), good thermal stability (≥60 °C), and excellent long cyclic stability [31,32,33]. Figure 2 shows that almost 71,000 research papers have already been published in the last ten years from 2010 to May 2021 (all data are recorded from “Scopus” up to 22nd of May 2021) [1,34,35,36]. The low ionic conductivity for room temperature ionic liquids (RTILs) is one of the major challenges for market applications. In this review, an overview of the recent research on different types of ILs electrolytes, their advantages and challenges for Li-ion battery (LIB) and supercapacitor (SC) applications has been discussed. In addition, the future scope of research on advanced ILs electrolytes for the next generation of EES devices is provided.

## 2. Types of Ionic Liquids (ILs)

Ionic liquids (ILs) are usually molten salts that possess a melting point below 100 °C. Most of these salts are organic salts with a large diversity of designability. Typically, these molten salts are composed of an asymmetric organic cation (pyridinium (PY), pyrrolidinium (PYR), imidazolium (Im), sulfonium, ammonium, etc.) (Figure 3), and a weak organic/inorganic anion (BF_4_^−^, PF_6_^−^ σ, triflate (CF_3_SO_3_^−^), bis(trifluoromethanesulfonyl imide) (TFSI) ((CF_3_SO_2_)_2_N^−^), etc.) combination [24,37]. Ionic liquids generally exhibit unique, remarkable properties such as very low vapor pressure, nonvolatility, high ionic conductivity (10^−3^–10^−2^ S cm^−1^), high electrochemical (up to 5 or 6 V vs. Li^+^/Li) and thermal stability with a large range of temperature [38]. The cations–anions combination can also be individualistically chosen to tune the physicochemical properties such as melting point, ionic conductivity, density, refractive index (r.i.), solubility, viscosity, etc., for ILs [39]. Due to these exceptional, unique properties, ILs currently attract a lot of advanced research as a new class of novel electrolytes for Li-ion batteries and supercapacitors [23].

Based on their chemical structure, ILs are mainly divided into three categories such as aprotic, protic and zwitterionic ILs (Figure 4) [40]. Aprotic and protic ILs both exhibit almost all the above-mentioned properties, but the main difference between these two ILs exists in the presence of a free proton or available proton on the cation of protic type ILs. Aprotic type ILs exhibit wide, stable electrochemical potential windows that make them suitable as electrolytes for supercapacitors and Li-ion batteries. Protic types ILs are used in fuel cells, and zwitterionic ILs are usually used as ILs-based membranes.

The properties of ILs as electrolytes for Li-ion batteries and supercapacitors plays an important role in regulating the overall electrochemical performance of energy storage systems (ESSs). RTILs have drawn a lot of efforts for ESSs due to their high ionic conductivity, non-flammability and wide potential windows. The tunability of groupings of cations–anions, as well as the chance of achieving changes of cation–anion combinations, extends access to targeted ILs. This has led to a significant expansion in the development of a number of new interesting ILs for the next generation ESSs. Based on cationic salt components, different types of ILs for energy storage device applications have been discussed below [36,41,42,43].

### 2.1. Imidazolium-Based ILs

Enormous research efforts are focused on imidazolium-based ILs as electrolytes for Li-ion batteries and supercapacitors owing to their advantages, such as tunable physicochemical properties, low viscosity and high ionic conductivity. Most of the imidazolium-based ILs such as 1-ethyl-3-methyl-imidazolium (EMIM^+^), 1-butyl-3-methyl-imidazolium (BMIM^+^), etc., are mainly non-amphilic.

It has been reported that the 3-butyl imidazole tetrafluoroborat (BMIMBF_4_) electrolyte exhibits a higher potential window (up to 3.5 V) and higher energy density for reduced graphene oxide and CMK-5 composite electrode-based SCs, compared to KOH (up to 1.0 V) and LiPF_6_ (up to 2.5 V) electrolytes [44]. Boujibar, O. et al. observed that the ethyl-methyl imidazolium tetrafluoroborate (EMIMBF_4_) electrolyte not only provides a wide potential window, up to 3.5 V with excellent electrochemical stability but also exhibits the highest specific capacitance (198.15 F/g), a high energy density of 82.93 Wh/kg as well as a power density of 3487 W/kg for activated (AC) based SCs [45]. Momodu, D. and co-workers obtained a 3.0 V stable potential window for (EMIM)(TFSI) ionic liquid-based symmetric SCs from Capsicum seed-porous carbon [46]. 1-ethyl-3-methylimidazolium bis(trifluoromethylsulfonyl)imide (EMIM)(TFSI)/ACN ILs offer wide potential windows up to 3.5 V with a specific capacitance of 43.5 F/g, an energy density of 74 Wh/kg and power density of 338,000 W/kg at 4.2 A/g of activated graphene-based SC electrodes (Kim, T. et al. [47]). Pham, D. T. et al. also achieved up to a 4.0 V potential window using BMIMBF_4_ electrolytes for carbon nanotube (CNT)-bridged graphene 3D building blocks for ultrafast compact supercapacitors, which exhibited 49 F/g (0.5 A/g) and an energy density of 110.6 Wh/kg and power density of 400,000 W/kg [48]. Que, M. et al. reported a safe and flexible electrolyte for Li-ion batteries. This electrolyte was prepared with a mixture of (EMIM)(TFSI) incorporated into 3P(MPBIm-TFSI), which exhibited thermally stable characteristics up to 370 °C and a stable potential window above 4.8 V. This solid-like composite electrolyte (SLCE) was non-flammable, had negligible leakage properties and high ionic conductivity of 1.2 mS cm^−1^ [49]. However, at low temperatures, the high viscosity and low electrical conductivity of imidazolium-based ILs hamper their large-scale industrial applications. To enhance the electrical conductivity of imidazolium-based ILs at low temperatures, the mixing of hydrophobic proton organic solvents has been discussed in many studies.

### 2.2. Pyrrolidinum-Based ILs

Pyrrolidinum-based RTILs have also been widely applied in SCs and Li-ion batteries as a solid-state electrolyte [50]. Substitution of the pyrrolidine cation improves the ionic conductivity of ILs-based electrolytes. *N*-butyl-*N*-methylpyrrolidine(trifluoromethyl sulfonyl)imide (Pyr_14_TFSI) hydrophobic ionic salt has drawn a lot of research interest for ESS applications owing to its excellent thermal and electrochemical stability (up to 5.5 V) at high temperatures (up to 300 °C) [51,52]. Yang, B. et al. examined the physicochemical properties of pyrrolidinum-based ILs, such as *N*-propyl-*N*-methylpyrrolidiniumbis (trifluoromethanesulfonyl)imide (Pyr_13_TFSI) electrolyte with organic additives and lithium bis(trifluoromethanesulfonyl)imide (LiTFSI) for high-safety Li-ion batteries [53]. This mixed electrolyte results in high thermal stability, non-flammability, low viscosity with a wide potential window of 4.8 V. This ILs-based electrolyte also offers low bulk resistance and the lowest interface resistance to Li-anode [54]. On the other hand, the porosity and electrochemical properties of carbon-based electrode materials can significantly influence the conductivity and polarization of the ILs for SCs device performance. Lazzari, M. et al. reported that the capacitance of AC electrodes and Pyr_14_ TFSI/(EMIM)(TFSI) electrolyte-based SCs were greatly influenced by the polarization of the cation parts of the ionic liquids on the surface of AC electrodes [55,56]. However, a lot of research work is going on all over the world on pyrrolidinum-based ILs for Li-ion and SC applications, but low-temperature conductivity and poor cycle life hinder its market application due to larger non-polar or surfactant ILs [50,57]. Quezada, D. et al. designed a safer Li-ion battery using zinc stannate as the anodic material and 1 M-LiNTF_2_ solution in 1-methyl-1-propylpyrrolidinium bis(trifluoromethanesulfonyl)imide ((MPPyr)(TFSI)) or 1-butyl-1-propylpyrrolidinium bis(trifluoromethanesulfonyl)imide ((BMPyr)(TFSI)) as ILs electrolytes. They observed that the Li-ion battery performance strongly depended on the cation structure of the ILs electrolytes, which offered discharge capacitance values of 306.3 mAh/g for BMPyrTFSI and 269.2 mAh/g for MPPyrTFSI [58]. Deb, D. et al. reported that hexafluoropropylene copolymer (P(VDF-HFP))-based “active” polymer membranes, prepared by pyrrolidinium ionic liquid-based nanofluid, improve the electrochemical potential windows up to approximately 5.3 V (vs. Li/Li^+^) for Li-ion batteries [22]. Kalinova, R. developed a novel pyrrolidinium-containing polymeric ILs (PILs) with a high conductivity suitable for composite supercapacitor electrodes. This 10–35 wt.% PIL in DMF electrolyte exhibits a stable capacitance, long cycling life of up to 7000 cycles with 87% capacitance retentions [59].

### 2.3. Quaternary Ammonium-Based ILs

In respect to electrochemical stability and wide potential window (>5 V vs. Li/Li^+^) for ESSs, quaternary ammonium-based ILs are one of the most promising electrolytes compared to imidazolium and pyrrolidinum-based ILs owing to their short chain. The most well-known quaternary ammonium-based ILs are *N*,*N*-diethyl-*N*-methyl-*N*-(2-methoxyethyl) ammonium TFSA (DEME-TFSA), and *N*,*N*-diethyl-*N*-methyl-*N*-(2-methoxyethyl) ammonium tetrafluoroborate (DEME-BF_4_) [60,61]. This DEME-BF_4_ ILs electrolyte can also enlarge the potential window (up to 6.0 V) and improve the electrochemical performance due to their high ionic conductivity (4.8 mS cm^−1^ at 25 °C) compared to other ILs such as EMIMBF_4_ [62]. Nevertheless, the larger cations size, high viscosity due to less Van der Waals interactions between anions and cation and low ionic conductivity limit their application for quaternary ammonium-based ILs [63].

### 2.4. Pyridinium-Based ILs

In comparison with other ILs, pyridinium-based ILs are also known as one of the novel ILs for EES devices, and extensive research on pyridinium-based ILs and their electrochemical activity, stability, reactivity, etc., is still required [64,65]. 1-butyl-3-methylpyridinium hydrogen sulfate ((BMPy)(HSO_4_)), 1-butyl-3-methylpyridinium ethylsulfate ((BMPy)(ESO_4_)), 1-butyl-3-methylpyridinium tetrafluoroborate ((BMPy)(BF_4_)), 1-butyl-3-methylpyridinium methylsulfate ((BMPy)(MSO_4_)), 1-butyl-3-methylpyridinium tetrafluoroborate ((BMPy)(BF_4_)), etc., are known as common pyridinium-based ILs, which have displayed remarkable performances in energy research [38,65,66].

### 2.5. Phosphonium-Based ILs

To date, phosphonium-based ILs as electrolytes in EES devices have not been extensively explored in large-scale applications due to their characteristically large cations and low ionic conductivities. However, this type of ILs could be an alternative electrolyte in the future as a combination with other ILs owing to their remarkable properties such as thermal stability (up to 400 °C in some cases) and fast electrochemical activity compared to imidazolium and ammonium-based ILs [38,67]. Table 1 represents few useful ionic liquid-based electrolytes for energy storage applications.

## 3. Ionic Liquids (ILs) for Li-ion Batteries

The roadmap to competitive alternative clean energy sources by 2050, stimulates new research on the development of advanced technologies in ESSs that can significantly reduce the emission of greenhouse gasses. Rechargeable Li-ion batteries (LIBs) are considered as the heart of this advanced technology, they have been in the market since 1991 due to their high energy density, up to 250 Wh/kg or up to 800 Wh/L. For their revolution in portable electronics, LIBs were recognized by the Nobel Prize in 2019 [7,79]. LIBs are being used every day in our daily lives, in portable/wearable electronics devices, hybrid vehicles, aerospace, ESSs grids, etc. [80,81]. Figure 5 represents the progress of the rechargeable batteries market as an energy storage system from 2005 to 2030. Around 2005, the battery technologies were dominated by lead-acid (Pb-acid) batteries, and they produced more than 80% of the energy (GWh) production. These Pb-acid batteries can still produce more than 60% of the energy in 2020 in a moderated form. It is expected that LIBs will become one of the most dominating technologies, by 2030, in energy production and will have the ability to produce more than 50% of energy (data were taken from Pillot, C. The Rechargeable Battery Market and Main Trends 2017–2030 (Avicenne Energy, 2019)) [80,82,83]. In LIBs, metal oxides are typically used as the cathode and graphite is mostly used as the anode. Nowadays, in order to improve the energy density of LIBs, huge efforts have been devoted to the development of high-capacity anodes and high voltage cathodes. In this case, the electrolyte is known as one of the most crucial elements for LIBs. To date, most of the electrolytes for LIBs are generally containing a mixture of Li-salt with organic carbonates such as diethyl carbonate (DEC), ethylene carbonate (EC), ethyl methyl carbonate (EMC), etc. [84,85]. However, these electrolytes can provide several advantages, such as good ionic conductivity, high capacity, etc., but their thermal instability, volatility and flammability impose a major safety issue for LIBs. For these reasons, ionic liquids (ILs) can be one of the best solutions for the next generation of LIBs by substituting flammable electrolytes and boosting battery safety. ILs not only ensure fast ion conduction, electrochemical stability with a wide potential window but can also overcome the safety concern issues coming from flammability, leakage possibility, and thermal instability of electrolytes. For more than 10 years, massive research has been going on ILs for LIBs throughout the world to address the above-mentioned challenges of LIBs and also to meet the global requirement [23,86]. Among different types of ILs 1-ethyl-3-methylimidazolium bis(trifluoromethylsulfonyl)imide (EMIM)(TFSI), *N*-butyl-*N*-methylpyrrolidiniumbis(trifluoromethylsulfonyl)imide (BMPyNTf_2_), *N*-propyl-*N*-methylpyrrolidiniumbis (trifluoromethanesulfonyl)imide (Pyr_13_TFSI), etc., have been tested for LIBs. A mixture of lithium bis(trifluoromethanesulfonyl)imide (LiTFSI) with any of these ILs could be used for LIBs to achieve thermal and electrochemical stability with wide potential windows. These ILs can also reduce the lithium dendrite formation and growth, which is one of the great obstacles for the scaling up of LIBs due to internal short-circuiting. Conversely, ILs electrolytes can play a vital role in the formation of a passive layer, known as solid electrolyte interphase (SEI) during lithiation/delithiation of Li-ions between graphite anode layers, which is a key point for battery life stability [87,88,89].

Rothermal, S. and co-workers reported a suitable ILs-based electrolyte mixture Pyr_14_TFSI-LiTFSI for “dual-graphite” electrode-based LIB cells. This IL-electrolyte exhibits not only high electrochemical stability but also form stable SEI at the graphite anode. The discharge capacity of this cell remarkably increased from 50 mAh/g to 97 mAh/g [90]. It was also observed that the mixture of two or more different anions could influence the lithiation/delithiation process of Li-ions in LIBs. Onset potentials of anion intercalation can be displayed as TFSI^−^/FSI^−^ (4.42 V vs. Li/Li^+^) < TFSI^−^ (4.44 V vs. Li/Li^+^) < FSI^−^/TFSI^−^ (4.46 V vs. Li/Li^+^) < FSI^−^ (4.53 V vs. Li/Li^+^) [87]. Balabajew, M. et al. also revealed that the electrolyte acts solely as an ion source and ion transport medium between the anode and cathode for LIBs. They studied the cell performance using Pyr_14_TFSI with different amounts of LiTFSI electrolyte and observed that the ionic conductivity (σ_dc_) of the electrolyte significantly decreased with the increase in mole fraction of LiTFSI (Figure 6) [91]. The decrease in ionic conductivity was mainly due to the formation of (Li(TFSI)_n_)^(n−1)−^ clusters, which enhanced long cycle-life, leading to the diffusion of Li^+^-ions collectively with the first coordination TFSI^-^-shell. As the number of (Li(TFSI)_n_)^(n−1)−^ clusters increased, there was an increase in the LiTFSI concentration, which also resulted in the decrease in the diffusion coefficient for Li^+^ and TFSI^−^. As a result, the viscosity of these mixtures’ electrolytes increased prominently, and ionic conductivity (σ_dc_) decreased [92,93]. Marczewski, M.J. et al. proposed a novel electrolyte concept for LIBs, known as the “ionic liquid-in-salt”. Their study on (1−x)EMIMTFSI: (x) LiTFSI, 0.66 ≤ x ≤ 0.97, clearly explained that at superior temperatures, the several dual-liquid and solid-phase regions had been characterized by a wide range of thermal stability, significant mechanical integrity and high ionic conductivity. They had the highest conductivity for the composition x = 0.70 and x = 0.75 (σ ≈ 6 × 10−3 S cm^−1^) of (1−x)(EMIM)(TFSI): (x) LiTFSI and are associated with the final melting point (at 138 °C and 151 °C, respectively) of the materials. Figure 7 represented that the overall high ionic conductivities are observed for 0.70 < x < 0.90 and lower conductivities for x > 0.90 for the mixtures [94].

In other work, Balo, L. et al. reported synthesis and characterization of gel polymer electrolyte (GPE) based on polymer polyethylene oxide (PEO) and LiTFSI/(EMIM)(TFSI) for high-performance lithium polymer batteries (LPBs). This ILs mixed electrolyte exhibits high thermal stability of up to 300 °C, high ionic conductivity (σ ≈ 2.08 × 10^−4^ S cm^−1^), a wide potential window (4.6 V) with high Li-transfer number (t_Li+_ = 0.39) [95]. Various research groups proposed that the addition of ILs as additives with organic electrolytes can enhance thermal stability, electrochemical stability, low viscosity and safety of LIBs. Chatterjee, K. et al. synthesized a non-flammable dicationic ionic liquid, 1,1′-(5,14-dioxo-4,6,13,15-tetraazaoctadecane-1,18-diyl) bis(3-(sec-butyl)-1H-imidazol-3-ium) bis((trifluoromethyl)-sulfonyl)imide as an electrolyte additive for LIB applications. They studied and compared the thermal, electrochemical and transport properties of the full cells made of the lithium nickel cobalt manganese oxide (NMC) cathode, graphite anode and ethylene carbonate (EC)-dimethyl carbonate (DMC) (1:1, *v*/*v* + LiPF_6_) mixture electrolytes with and without this IL as an additive. Figure 8a represents 100 cycles of galvanostatic charge–discharge (GCD) curves at a constant current rate of 10 mA/g for both types of coin cells with and without IL additive. These cyclic performances clearly reveal that the cells with the IL additive performed for 6 days more, for the same number of cycles, compared to the cells without the IL additive. Capacitance retention curves (Figure 8b) also show that the discharge capacity of the cell without IL additive was a little higher than IL additive cells for the first few cycles and after that (around 30 cycles), the IL additive cells performed better (retains around 73% of capacity), which signifies that the addition of IL in electrolytes enhance the electrochemical performance of the electrolytes and cells for long cycle life. The electrochemical impedance (EIS) curves (Figure 8c) for both cells with and without the IL additive after 100 cycles also confirms that the interface impedance of the full cell with the IL additive is very low compared to without the IL additive electrolytes, which means the addition of the IL additive developed the SEI layer formation on the graphite anode electrode. The flammability test (Figure 8d–f) was also carried out to check the safety issue for these electrolytes to be used in LIBs. It was observed that this pure IL additive did not catch fire, while a conventional electrolyte catches fire immediately, and 20 mM IL additive electrolyte catches fire slowly after 15 s. This experiment strongly recommends that this IL additive electrolyte is safer than market-available conventional organic electrolytes [96]. However, the rapidly growing demand for high performance, high energy-power capacity, reliability, safety, as well as a low cost cannot be fulfilled with existing LIBs technology. There are a lot of developments in advanced LIBs and ILs-based electrolytes that need to be done. New ideas and research in the development of electrolytes that can operate at a wide range of temperatures, wide electrochemical potential windows, long cycle life, etc., should be promoted in future LIBs to reach scalability, sustainability, manufacturability and recyclability for next-generation ESS.

To forestall and fulfill the rapidly growing energy demands, the development of advanced EES device technologies beyond LIBs chemistry is also enormously important for large-scale energy applications. These “post-lithium” battery technologies, which are essentially based on a high raw material abundance and low-cost metal-ion batteries such as sodium-ion (Na-ion), potassium-ion (K-ion), magnesium-ion (Mg-ion), aluminum-ion (Al-ion), zinc-ion (Zn-ion), etc., have a lot of advantages. However, their developments are still in the early stages of research when compared to LIBs [82,97,98,99]. One of the key challenges to all these metal-ion batteries is to develop a high-performance, safe and reliable electrolyte. From the perspective of different types of electrolytes, ILs-based electrolytes are promising and attractive for developing novel electrolytes for these metal-ion batteries owing to their various unique properties over commercial electrolytes [99,100,101]. A short discussion on these metal-ion batteries over “post-lithium” battery technologies has been discussed here.

Among different types of metal-ion batteries, sodium-ion batteries (SIBs) are predicted to meet sustainable requirements and performance in energy storage applications. Compared to LIBs, sodium has a lot of advantages due to higher abundance, about 24,000 ppm (20 ppm for Li), low raw material (Na_2_CO_3_) cost (around 100 times lower than Li_2_CO_3_) and wider distribution. Na-metal also offers higher battery voltage due to a low redox potential (2.71 V vs. SHE). The larger ionic radius of the Na^+^ ion than Li^+^ ion significantly reduces solvation in polar solvents, which can control the migration of Na metal at the interface and promotes the intercalation of the Na^+^ ion. However, the Na-ion being heavier than the Li-ion subsequently causes an estimated capacity loss, which can be overcome by replacing the heavy copper current collector with lighter aluminum. Electrolytes play an essential role for SIBs owing to their fundamental difference from LIBs in physical and chemical nature related to Na-ions. Over the past few years, a lot of great research efforts have been made on an electrolyte to find the suitable combination of electrolytes for SIBs [102,103,104,105]. In most of the research, organic carbonate-based electrolytes with sodium salts (such as NaClO_4_, NaPF_6_, etc.) are considered as one of the most promising electrolytes for SIBs. However, flammability, volatility, high thermal sensitivity, etc., of such kinds of organic electrolytes limit their commercial application in emerging SIBs. There is limited research having been implemented on ILs-based electrolytes in SIBs, and therefore, urgent development of ILs-based high-performance and safe electrolytes is required for the next-generation SIBs. Up to date, numerous cathode materials (such as NaVOPO_4_, Na_x_TMO_2_, Na_1.56_Fe_1.22_P_2_O_7_, NaCrO_2_, Na_2_ZrCl_6_, Na_0.45_Ni_0.22_Co_0.11_Mn_0.66_O_2_, NaFeP_2_O_7_, NaFe_0.4_Ni_0.3_Ti_0.4_O_2_, etc.) are studied in imidazolium, pyrrolidinium, ammonium, and the anions of TFSI^−^, FSI^−^, and BF_4_ ILs electrolytes for SIBs [99,100,106,107]. Giffin, G. A. clearly explained in her review article on “beyond lithium” battery technologies that some layered-structure metal-oxides exhibit better cycling stability in ILs electrolytes at room temperature than conventional organic electrolytes due to the reduced solubility of transition metals (such as Mn) in ILs [100]. Quan, P. et al. developed a novel electrolyte for SIBs using a mixture of sodium bis(trifluoromethane sulfonyl)imide (NaTFSI), *N*-butyl-*N*-methylpyrrolidinium bis(trifluoro-methanesulfonyl)imide (Py_14_TFSI) as co-solvent and commercial carbonate solvents, i.e., EC-PC (1:1), EC-DMC (1:1), and EC-PC-DMC (3:1:1). These ILs-based mixture electrolytes significantly improved the ionic conductivity and the electrochemical potential window of the SIBs [102]. The implementation of ILs-based electrolytes in SIBs proposes that the use of ILs can enhance the electrochemical performance of SIBs that beat those of conventional organic electrolytes. However, significant work is still needed so that they can compete with organic electrolytes in next-generation SIBs [39,99,104].

Recently, similar to Na-ion batteries, potassium-ion batteries (PIBs) also have attracted a massive research interest as an alternative to LIBs owing to their abundant reserves, low cost, the fast migration rate of K^+^ and expected higher operating potential. The PIBs can offer higher energy density due to the lower standard electrode potential of K (−2.93 V) compared to Li (−3.04 V), which makes them suitable in high voltage outputs for next-generation EESs. However, PIBs still suffer from low Coulombic efficiency and unstable solid–liquid interfaces due to side reactions between K-electrodes and electrolytes, which can be figured out by choosing proper electrolytes. Among different types, K-salt-based imides (KFSI, KTFSI, etc.) electrolytes, the KFSI-based electrolyte has the ability to form a stable SEI layer that can significantly improve the long lifetime and stable cycling performance. Yamamoto, T. et al. proposed a high-voltage PIB (5.72 V) with an ionic conductivity of 4.8 mS/cm using (PY_13_)FSI as the electrolyte at 25 °C [108,109]. Fiore, M. and co-workers obtained a high capacity (119 mAh/g), 87.4% capacity retention at 0.1 C after 100 cycles by combining potassium manganese hexacyanoferrate (KMF) cathode, graphite anode and the KFSI-(PY_13_)FSI ionic liquids as an electrolyte. This ILs-based electrolyte also provides 99.3% Coulombic efficiency due to the high electrochemical stability of KFSI-(PY_13_)FSI electrolyte under high potential [110]. X. Liu et al. also reported that the interaction between K^+^ cation and TFSI^−^ anion has suitable compatibility with a lighter aluminum (Al) current collector in PIBs at a high salt concentration [111]. However, PIBs have a number of advantages that make them attractive as an alternative for the next-generation batteries, but studies on high capacity, high energy density, stable cathode materials and suitable electrolytes are still in the primary stages of research. Currently, there is less research on ILs electrolytes in PIBs, and new research needs to be focused on the optimization of ILs-electrolytes in developing advanced PIBs [112].

Nowadays, Magnesium-ion batteries (MIBs) have also attracted a lot of research interest as one of the “post-lithium” battery technologies due to their high volumetric energy density (3833 mAh/cm^3^), excellent safety for no dendrite effect and low cost of raw materials. However, many problems of MIBs still need to be addressed before commercialization, including the development of electrochemically stable electrodes, suitable new electrolytes, low voltage operation problems, etc. Many research groups have already proposed that conventional electrolytes (such as Mg(TFSI)_2_, Mg(ClO_4_)_2_, Mg(PF_6_)_2_, etc.) in carbonate or ether solvents perform poorly in MIBs because of SEI layer formation on the Mg-Anode surface. On the other hand, higher charges of Mg^2+^ and a strong binding force with cathodes in MIBs cause the ruin of cathode material structures and poor cycling performance due to a slower shuttle rate between anodes and cathodes [99,112]. Lei, X. et al. reported a Mg-ion-based dual-ion battery, using 3,4,9,10-perylenetetracarboxylic diimide (PTCDI) as the organic anode, expanded graphite (EG) as the cathode, and (0.4 m Mg(TFSI)_2_-(PY_14_)TFSI) ionic liquid as the electrolyte. This PTCDI showed good solubility, good structural stability and offered a discharge capacity of 57.7 mAh/g at 2 °C in the potential window of 1–4 V with capacity retention of 95.7% after 500 cycles at 5 °C [98]. This research also demonstrated that IL electrolytes could be very useful for MIBs in the future to overcome the common problems with conventional electrolytes [112].

Among ever-growing emerging EES technologies for electrical energy storage, Zn-ion batteries (ZIBs) have received increasing attention because of their low cost, high abundance, high eco-efficiency, high safety, low potential (−0.762 V vs. SHE) and high theoretical capacity (820 mAh/g) [58,113,114,115]. However, finding a suitable cathode material and electrolyte combination is still a great challenge for researchers of the recent ZIBs technologies. To date, Zn is suffering from corrosion effects, surface passivation and the growth of dendrite after long cycles in common alkaline electrolytes, which significantly limits the battery performance of ZIBs. Therefore, nonaqueous ILs-based electrolytes are more effective than acidic or neutral electrolytes to solve the problems of low Coulombic efficiency caused by the side reactions of Zn, unwanted hydrogen evolution reactions (HER) and the dendrite formation of ZIBs [112]. Consequently, ionic liquid electrolytes based on PVDF, HFP, PEO, etc., as electrolyte matrixes and ILs additives are proposed as one of the promising electrolytes in ZIBs. Recently, Ma, L. and co-workers fabricated a ZIB using (EMIm)BF4-Zn(BF_4_)_2_ as the electrolyte and cobalt hexacyanoferrate (CoHCF) as the cathode materials. This ZIB with IL-based electrolytes exhibited excellent electrochemical stability with 98% capacitance retention over 40,000 cycles, high Coulombic efficiency (around 100%), high ionic conductivity and solved the problem of HER [116].

Among other metal-ion batteries, Aluminum-ion batteries (AIBs) also possess a lot of advantages such as being lightweight, three-electron transfer electrode ability (Al^3+^ + 3e^−^

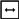
Al), low cost, most abundant and most significantly high volumetric capacity (8040 mAh/g), and high gravimetric capacity (2980 mAh/g), just in the second position next to LIBs (3870 mAh/g) [99,112]. However, many studies are still ongoing, and state of the art of AIBs, carbon-based electrodes and ILs electrolytes combinations have proved to be one of the most suitable combinations due to their synergistic effect and many research groups also focus on AlCl_3_-type ILs electrolytes in AIBs [117,118]. Furthermore, inorganic material (e.g., V_2_O_5_) cathodes and ILs electrolytes such as AlCl_3_/(EMIm)Cl can be useful in AIBs. However, to address the corrosive effect of AlCl_3_-type ILs to Al foil, different combinations of ILs such as AlCl_3_/(BMIm)Cl, Al(TfO)_3_/(BMIm)TfO, etc., have been examined to develop the advantages of AIBs in real-life applications over LIBs. From a practical point of view, the progress of AIBs is still lagging due to challenges and difficulties in finding high-performance electrode materials and electrolytes [99,100,112,119].

## 4. Ionic Liquids (ILs) for Supercapacitors (SCs)

Supercapacitors (SCs) can store energy via redox reaction or the formation of electric double layers (EDLs) of ionic species over the surface of electrodes at the electrode/electrolytes interface. They can reach the global requirements for ESS due to rapid power output, fast charge–discharge ability and longer lifetime than batteries [4,120,121,122,123]. SCs can be independently used as an alternative renewable energy source in the place of batteries in many fields such as wearable electronics, electric buses, laptops, mobiles, aerospace and various diagnostic instruments in medical systems, which boosts the growing interests in the field of advanced SCs research. The energy density (E_a_) is one of the key points that can evaluate the SCs. The low energy density of SCs obstructs its large-scale industrialization. The energy density of SCs is related to voltage window (V), and specific capacitance (C) of the device (E_a_ = ½ CV^2^) [4,6]. As energy density is directly proportional to the square of potential window (V^2^), then significant improvement can be accomplished by developing new high-capacity electrode materials and/or widening the electrochemical potential window of the SC device. The electrochemical potential window (EPW) of an SC is generally determined by the electrochemical stability of the electrolyte used in the device [1,124,125]. In most cases, the EPW cannot reach higher than 1.0 V for aqueous electrolytes due to the decomposition of water molecules [126,127]. For organic electrolytes with higher decomposition potential (such as propylene carbonate (PPC), acetonitrile (ACN), etc.), cell voltage can reach up to 2.7 V, but unfortunately, they suffer from toxicity, inflammability and major safety issues [14,128]. Thus, in developing safe and wide potential electrolytes, compared to aqueous and organic electrolytes, ionic liquids (ILs) have gained a lot of widespread interest in recent years as SCs electrolytes, owing to their many merits such as discrete anion–cation combination, negligible volatility, non-flammability, wide potential window (up to 4.5 V) and with a high ionic conductivity [23,32,33]. A safe, wide range of operational temperatures can also be attained for SCs using ILs as electrolytes. Balducci, A. and co-workers demonstrated and achieved a wide potential window of up to 3.5 V using activated carbon (AC) as the negative electrode, polymethylthiophene as the positive electrode and pyrrolidinium-based IL as the electrolyte. Still, the high viscosity and low ionic conductivity below room temperature of this electrolyte limited the operational temperatures below 50 °C for this SC device [51,129]. To overcome high viscosity and low conductivity challenges for ILs, it was observed that a suitable nanostructured combination of CNT or graphene-based electrodes and a mixture of ILs electrolytes in an SC could significantly enlarge the operational temperature range and ionic conductivity above room temperature [130]. Lin, R. et al. revealed that the capacitive energy storage of SCs could be increased from −50 °C to 100 °C (Figure 9a) using ILs electrolytes. The change of capacitance vs. temperature (Figure 9b) for onion-like carbon (OLC) and vertically aligned carbon nanotube array (VA-CNT) electrodes shows that the temperature range was restricted to 110 °C using 1 M tetraethylammonium tetrafluoroborate (TEA-BF_4_) in PC. This was mainly for the large capacitance drop at low temperatures and for electrolyte oxidation beyond 2.5 V at high temperatures, which was one of the major challenges for commercial SCs. However, the mixture of *N*-methyl-*N*-propylpiperidinium bis(fluorosulfonyl)imide (PIP_13_FSI) and *N*-butyl-*N*-methylpyrrolidinium bis(fluorosulfonyl)imide (PYR_14_FSI) in conjugation with (PIP_13_FSI)_0.5_(PYR_14_FSI)_0.5_ enhances the wide temperature window of 150 °C [73,130].

To date, many research groups carried out their extensive research on non-amphilic IL electrolytes such as imidazolium and pyrrolidinium-based ILs for supercapacitive application due to their suitable high ionic conductivity and relatively lower viscosity. Imidazolium-based ILs usually exhibit high ionic conductivity compared to pyrrolidinium-based ILs, which possess wide EPWs. Mao, X. et al. discovered surface-active ILs (SAILs) containing self-assembly amphiphilic structures for EDLCs. These SAILs offered high capacitive performance at electrified surfaces and exhibited unusual interfacial ion distributions due to Van der Waals interactions [131]. In another work, Ray, A. et al. reported a PPC polymer embedded (EMIM)(TFSI) ILs-based electrolyte for direct sputter-grown Mn/MnO_x_@Graphite-Foil electrodes of flexible all-solid supercapacitors. This PPC embedded ILs exhibited a wide electrochemical stable potential window of up to 2.2 V with a long cycle life stability of up to 5000 cycles for this flexible needle-like Mn/MnO_x_ nanostructured on the surface of graphite foil [120]. In the case of the increased energy density of SCs, the shape and morphology of the electrodes play an important role for the electrolyte. Porous carbon-based electrodes are mostly used in SCs compared to planar electrodes to achieve high specific capacitance as well as high energy density due to high accessible surface area for electrolytes. Most of the ILs produced a layered structure on the surface of planar electrodes that changed the polarity of the electrode owing to the asymmetric nature of electrolyte ions. ILs can form a monolayer of ions inside the pores of a nanoporous electrode and a multilayer inside wider pores. It was also observed that narrow pores could provide higher energy density at lower EPWs, and wider pores can exhibit higher energy density at higher EPWs [52,132]. Burt, R. and co-workers also suggested that the higher concentration of pure IL-ions sometimes could not significantly increase the capacitance of nanoporous carbon electrode-based SCs due to difficulties in separating opposite charges of pure ILs [133]. Some research groups reported that many pseudocapacitors (MnO_2_, CuO, RuO_2_, etc.) with ILs suffer from a large voltage drop (IR-drop) due to relatively high intrinsic charge transfer resistance (R_ct_). In general, the electronic conductivity of such metal oxides is comparatively low compared to carbon electrodes and conventional electrolytes. The large cations of ILs cannot frequently disperse within the pores of electrodes and only adsorb on the surface of the electrode without infiltration into the lattice structure. Navathe, G. J. et al. fabricated nanostructured copper oxide (CuO) on a stainless steel substrate electrode and studied supercapacitive performance using 0.1 M 3-(1′-hydroxypropyl)-1-methylimidazolium chloride (HPMIM)(Cl) ionic liquid as the electrolyte. This electrode achieved a maximum specific capacitance of only 60 F/g at 10 mV/s [134]. However, ILs have tremendous potential in the field of energy storage applications, but the cost issue of ILs, unfortunately, is still a big challenge in the scalable commercial application due to the high purification cost. Upgraded knowledge and optimization in the application of ILs in SCs are required towards nanostructured electrodes, choice of anion–cation combinations, and the orientation of ions on the electrode surface. These can strongly influence the EPWs and the capacity of SCs.

## 5. Challenges and Future Motivation for ILs

According to the previous research works, ionic liquids (ILs) exhibit tremendous potential in the field of energy storage applications, mainly in LIBs and SCs. ILs are considerably different from conventional organic and aqueous electrolytes. Still, there are some problems in the energy storage mechanism of ILs in LIBs and SCs need to be further addressed (Figure 10). Typical ILs facilitate the widened electrochemical potential windows of LIBs and SCs but sometimes are unable to improve the capacitance due to ions separation difficulties, and this could be one of the crucial objectives for future research. Conversely, there is a fast-growing interest in ESSs in the automotive industry to develop an advanced device that can be operated at a wide range of temperatures as low as −60 °C for military purposes. The degradation of device performance at temperatures below −60 °C for most of the batteries is required to improve. The increase in viscosity of most ILs at lower temperatures deteriorates the charge transportability of the device, and unfortunately, there are very few ILs that can be worked at extremely low temperatures. The fundamental knowledge of ILs as electrolytes needs to be increased for the development of wide temperature, low viscosity, safe and high-voltage ILs for energy storage. A mixture of different types of ILs or the use of ILs as additives with electrolytes, which can reduce the viscosity at low temperatures, could be one solution to migrate the low-temperature problem for LIBs. The mixing of ILs with organic electrolytes can increase the ionic conductivity and operational temperature range for ESSs, but sometimes it can reduce the potential window as well as energy density for LIBs and SCs. Eutectic IL mixtures can be a promising candidate as electrolytes for advanced ESSs over a wide range of operational temperatures owing to their good electrochemical stability and lower viscosity. Newell, R.’s group, discovered a novel eutectic IL mixture of 1-propyl-3-methylpyrrolidinium bis(trifluoromethylsulfonyl)imide with (EMIM)(TFSI), which offered a wide, stable potential of 3.5 V at a wide temperature range (−70 °C to 80 °C) and was favorable for SCs application [67,135].

Although eutectic IL mixture electrolytes provide better charge storage performance, advanced technologies and optimizations are highly desirable to adjust the electrochemical properties of this mixtures’ electrolytes compared to pure ILs without sacrificing any desirable performance of energy storage devices.

Nowadays, ionogel electrolytes based on poly(ionic liquids) (PILs) and conventional polymers could be alternative electrolytes, which can dominate in future energy storage due to their safe high-voltage applications and self-healing properties [38,42]. Ionogel electrolytes based on various polymers such as poly(vinylidene fluoride) (PVDF), poly(ethyl oxide) (PEO), and poly(vinylidene fluoride-co-hexafluoropropylene) (PVDF-HFP) have been widely investigated for energy storage devices [49,73]. Innovative research and ideas are needed to improve the poor mechanical stability and regulating the self-healing nature of ionogel electrolytes for next-generation LIBs and SCs. There are a lot of major projects going on under the battery roadmap of the European Union (EU) for next-generation LIBs. One of the EU-funded (Battery 2030) projects entitled “Autonomous Polymer-based Self-Healing Components for high performant Li-Ion Batteries” or “BAT4EVER” is mainly focused on the self-healing mechanism of novel non-flammable polymerized ILs electrolytes, self-healing surface layer protected silicon (Si) anodes, and advanced core/shell structured NMC nanoparticles cathode for LIBs applications that will be substantial in strengthening the European battery industry to be competitive in the European and world battery market and in offering the European society with safer and long-lasting battery products.

## 6. Conclusions

In this short review, we have tried to highlight the recent progress in ionic liquids (ILs) electrolytes for energy storage, mainly for Li-ion batteries (LIBs) and supercapacitors (SCs). It has been observed that the wide electrochemical potential window, very low volatility, good thermal and electrochemical stability of ILs make them suitable as an alternative for the next generation of electrolytes in energy storage devices. The unique properties of the capability to perform at a wide range of temperatures in ILs have attracted huge interest for use in hybrid electric vehicles. However, most of the results explained in this review are still at the laboratory research scale for further development, and a lot of work on ILs-based electrolytes is yet to be completed before commercialization. ILs’ are still suffering from poor power performance compared to commercial organic electrolytes. Despite the disadvantages, the industrial application of ILs as high-voltage electrolytes for energy applications persists. We hope that this short review will inspire young researchers, students and readers and give them a brief introduction to ILs for further improvement.

## Figures and Tables

**Figure 1 materials-14-02942-f001:**
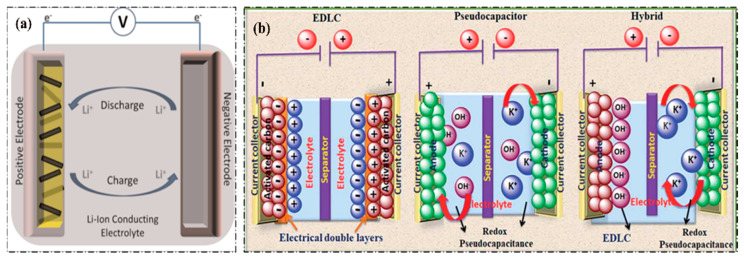
The charge storage mechanism of (**a**) Li-ion batteries (LIBs) and (**b**) different types of supercapacitors (SCs), (**a**) Reprinted with permission from Ref. [23]. Copyright 2017 American Chemical Society; (**b**) Reprinted with permission from Ref. [24]. Copyright 2019 The Royal Society of Chemistry.

**Figure 2 materials-14-02942-f002:**
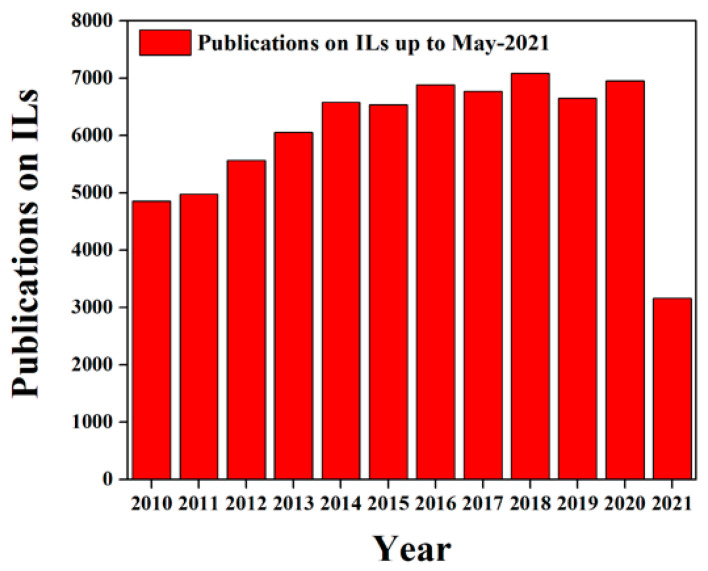
The number of publications in the area of ILs from 2010 to 2021 (All data are recorded from “Scopus” up to 22nd of May 2021).

**Figure 3 materials-14-02942-f003:**
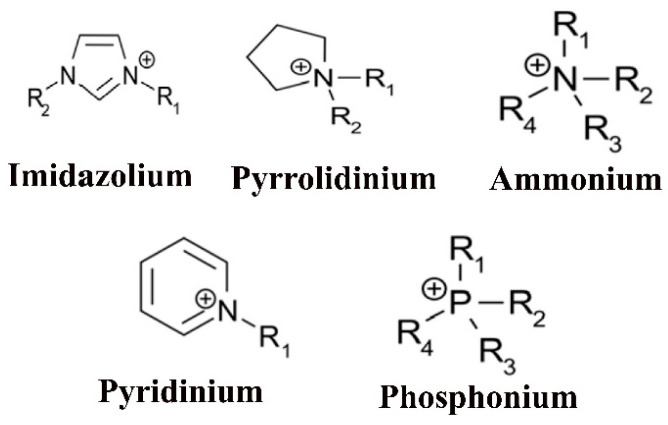
Different types of cations in ILs.

**Figure 4 materials-14-02942-f004:**
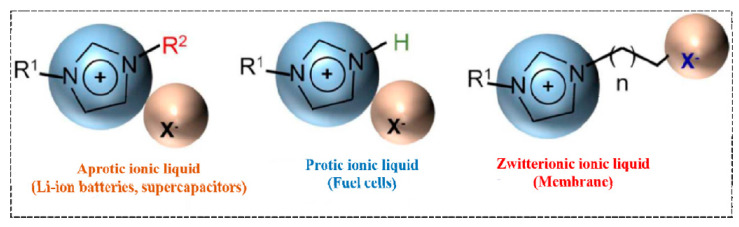
Based on composition, different types of ILs and their specific application, copyright; Reprinted with permission from Ref. [40]. Copyright 2009 Nature Materials. “Source: U.S. National Library of Medicine”.

**Figure 5 materials-14-02942-f005:**
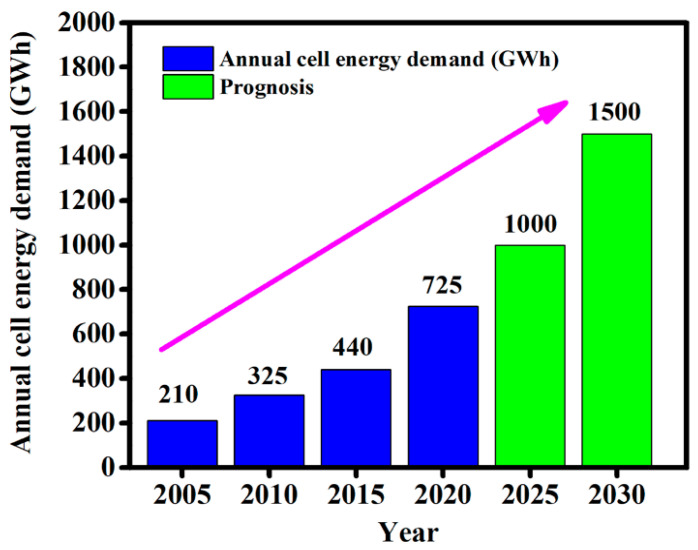
Progress of the rechargeable batteries market as an energy storage system from 2005 to 2030; data recorded from “The Rechargeable Battery Market and Main Trends 2017–2030 (Avicenne Energy, 2019)”.

**Figure 6 materials-14-02942-f006:**
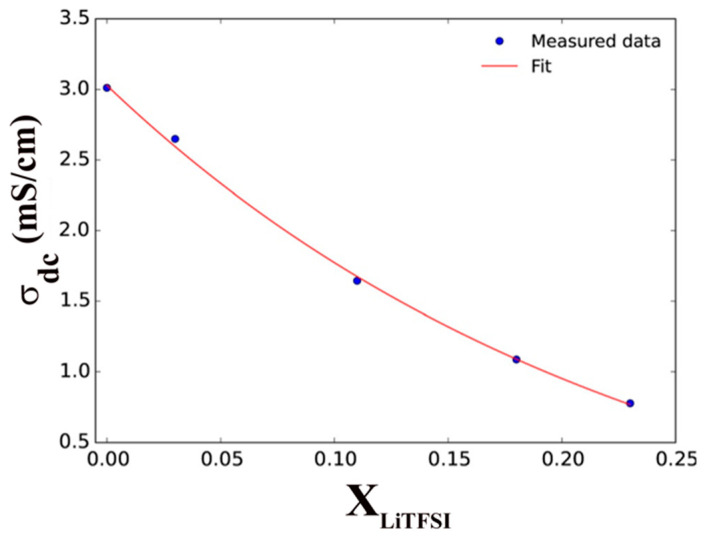
Variation of the ionic conductivities of different solutions of electrolytes with mole fraction of LiTFSI (X_LiTFSI_) in Pyr_14_TFSI, copyright; Reprinted with permission from Ref. [91]. Copyright 2015 Wiley-VCH Verlag GmbH. Reproduced with permission.

**Figure 7 materials-14-02942-f007:**
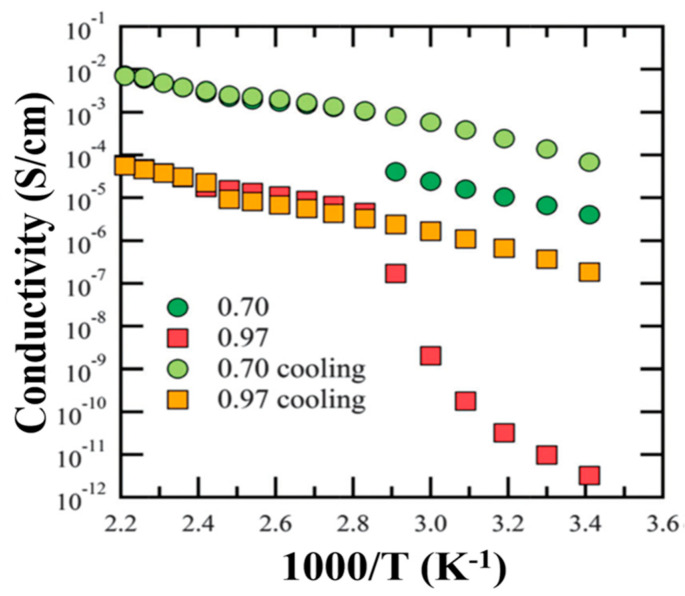
Comparison of ionic conductivities obtained for the Li_x_EMIM_(1-x)_TFSI (x = 0.70 and x = 0.97) system, data collected during heating and cooling, copyright; Reprinted with permission from Ref. [94]. Copyright 2014 The Royal Society of Chemistry.

**Figure 8 materials-14-02942-f008:**
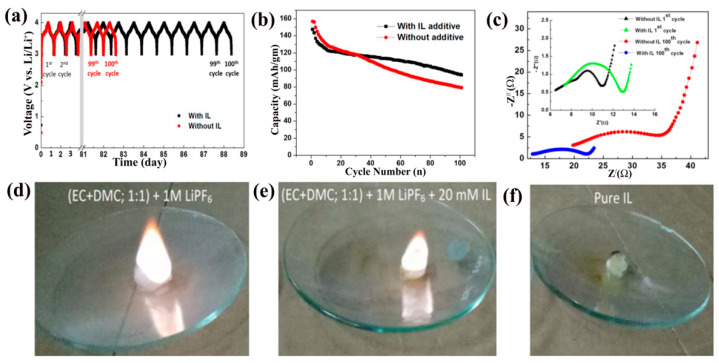
(**a**) Galvanostatic charging–discharging (GCD) at a constant current (10 mA/g) of the cells with IL (black color) as additive and without IL (red color). The GCD cycles were interrupted when the cell voltage exceeded 4.0 V or dropped below 3.0, respectively. Notice the break plot from 3 to 81 days, (**b**) discharge capacitance performances of the cell in electrolyte without (red) and with IL (black), (**c**) electrochemical impedance response after the 1st and 100th cycle of the cell for electrolyte without and with IL additive, flammability test for (**d**) conventional electrolyte, (**e**) conventional electrolyte with IL, (**f**) and pure IL, copyright; Reprinted with permission from Ref. [96]. Copyright 2020 Springer Nature.

**Figure 9 materials-14-02942-f009:**
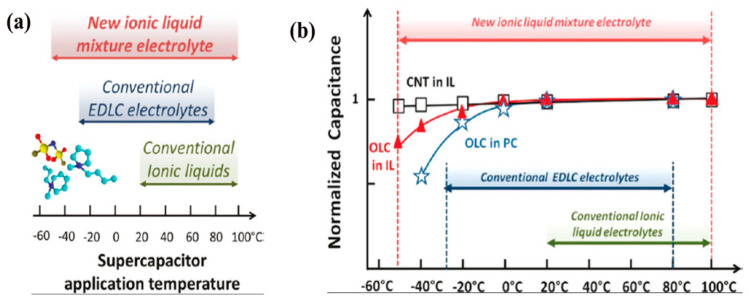
(**a**,**b**) Change of capacitance (C/C20 °C) versus temperature for the OLC and VA-CNT electrodes in (PIP_13_FSI)_0.5_(PYR_14_FSI)_0.5_ IL mixture and PC + 1 M TEA-BF_4_ electrolytes. Capacitances were calculated at 100 mV/s, except for the −50 °C (1 mV/s) and −40 °C (5 mV/s) experiments. This plot (**b**) shows that the use of the IL mixture extends the temperature range for SCs into the −50 °C to 100 °C range while conventional electrolytes using PC as the solvent are limited to the −30 °C to 80 °C range. C _20 °C_ was 80 mF and 4 mF, respectively, for OLC and VA-CNT cells. The potential window was 0 to 2.8 V, copyright; Reprinted with permission from Ref. [130]. Copyright 2011 American Chemical Society.

**Figure 10 materials-14-02942-f010:**
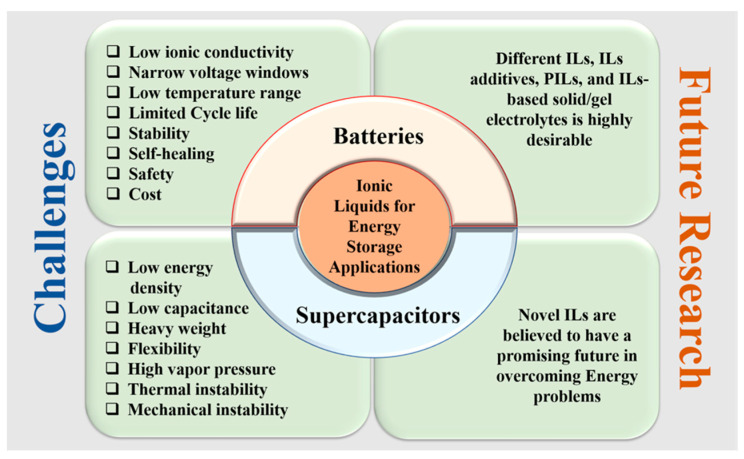
Challenges and future research on ILs for energy storage applications.

**Table 1 materials-14-02942-t001:** Few important IL-based electrolytes for EES applications are reported in the literature.

Ionic Liquid (IL)	Abbreviation	Viscosity (mPa s)	Conductivity (σ) (mS/cm)	Electrochemical Potential Window (V)	Ref.
Ethyl-3-methylimidazolium bis-trifluoromethylsulfonylimide	(EMIM)(TFSI)	34	10.8	4.0	[68]
1-ethyl-3-methyl imidazolium tetrafluoroborate	(EMIM)(BF_4_)	32	16.01	4.0	[69]
1-ethyl-3-methylimidazolium bis(fluorosulfonyl)imide	(EMIM)(FSI)	19	17.74	3.5	[70]
1-ethyl-3-methylimidazolium trifluoromethanesulfonate	(EMIM)(TfO)	45.7	8.5	4.1	[71]
1-butyl-1-methylpyrrolidinium bis(trifluoromethylsulfonyl)imide	PYR_14_TFSI	85	2.2	>4.8	[72]
*N*-propyl-*N*-methylpyrrolidinium bis(trifluoromethanesulfonyl)imide	PYR_13_TFSI	58.7	4.92	5.0	[73]
1-ethyl-3-methylimidazolium chloride	(EMIM)(Cl)	-	1.85	2.8	[74]
*N*-methyl-*N*-propylpiperidinium bis(trifluoromethylsulfonyl)imide	Pip_13_TFSI	-	10	4.35	[75]
*N*,*N*-diethyl-*N*-methyl-*N*-(2-methoxyethyl) ammonium tetrafluoroborate	DEME-BF_4_	-	4.8	6.0	[76]
1-ethyl-3-methylimidazolium hexafluorophosphate	(EMIM)(PF_6_)	-	1.93	3.2	[77]
1-butyl-3-methylimidazolium hexafluorophosphate	(BMIM)(PF_6_)	-	2.1	3.0	[78]

## Data Availability

Data sharing is not applicable.

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
