# Peer review of "Application of Ionic Liquids for Batteries and Supercapacitors"

_materials, 2021, doi:10.3390/ma14112942_

Round 1
Reviewer 1 Report
The manuscript elaborates the utilization of ionic liquids for LIBs and capacitors. It’s a hot topic, and deserves careful consideration. The manuscript is written well, although the current discussions are not sufficient for the final publication, and it certainly needs additional discussions to be acceptable in Materials. I would like to highly recommend this manuscript for publication in Materials. However, the authors need to provide the following modifications before the final acceptance:
1- More sophisticated figures/illustrations should be provided during revisions. The current illustrations will barely attract the attention of future readers. Please concentrate on the state-of-the-arts published in the last 2-3 years.
2- I recommend the authors update the title as: “Applications of Ionic Liquids for Metal-ion Batteries and Supercapacitors”
3- Please extend your discussions for the employment of ILs for other metal-ion batteries, specifically it’s urgent talking on Na-ion and K-ion batteries. Other suggestions could be Mg-ion, Zn-ion, and Al-ion batteries to have fruitful discussions for the future readers.
4- Please update Figure 2 with the published papers in 2021.
5- Please provide a comprehensive table regarding the physical and chemical properties of various ILs. For example, the viscosity, conductivity, concentration, electrolyte components, and electrochemical performance in various energy storage devices could be highlighted.
6- The current literature references do not fully cover the most recent highlighted research efforts in this field. Please provide additional refs and properly address the concerns, challenges, and progress on the efficient utilization of ILs in energy storage devices (mainly metal-ion batteries and capacitors).
7- Please design and present a concluding figure based on the current challenges and your suggested future research directions.
8- References need careful reconsideration and corrections. For example, authors should be sure about the accuracy and completeness of Refs. 14, 25, 39, 41, 51, 55, 57, 58, 66, 70, 71, 74
Author Response
Please find the respond to the comments of the Reviewer 1 in the attached file.

Reviewer 2 Report
The manuscript, “Application of Ionic Liquids for Batteries and Supercapacitors” reviews important ionic liquid families and candidates that have been used in batteries and supercapacitors. The manuscript manages to succinctly cover most systems where ionic liquids have been particularly successful in replacing or substituting conventional, carbonate-based solvents. This mini review will certainly be useful for the batteries/supercapacitors’ community, and would be suitable for publication in Materials, but there are a few additions and revisions I would like to suggest to the authors before publishing. They are as follows:
1. Although this is a short review and not meant to cover all areas in batteries, I believe that the usefulness of ionic liquids in beyond-lithium-ion and multivalent battery systems are worth mentioning. These multivalent metal-ion systems do not work very well with conventional electrolyte solvents and so, ionic liquids have been greatly helpful in realizing meaningful performances in these systems during the early days. So, I would recommend that the authors include a short discussion on this subject. There are few prior articles in the literature that cover this in detail, such as J. Mater. Chem. A, 2016, 4, 13378, Energy Storage Materials, 2019, 21, 136-153, and Zhu et al., 2021 (DOI: 10.34133/2021/9204217). Discussing these systems briefly and including these/other references would provide further reading material for interested readers.
2. There have been some important developments such as high-voltage stability, Li-metal stability, higher transference number etc. that were made possible by pyrrolidinium-based ionic liquids in lithium-ion, lithium-metal batteries, and multivalent-ion batteries. I would recommend the authors to expand the manuscript’s existing section on pyrrolidinium ionic liquids for lithium-ion / multivalent-ion batteries and highlight these developments.
Author Response
Please find the respond to the comments of the Reviewer 2 in the attached file.

Reviewer 3 Report
This review "Application of Ionic Liquids for Batteries and Supercapacitors" shows the research progress on the ionic liquids as the electrolyte for lithium-ion batteries and supercapacitors applications, pros and cons for different types of electrolytes, and research examples of ILs to show its advantages (e.g. wide potential window, safety) and potential issue (e.g. ion separation). It is a short but relatively comprehensive review for providing a basic understanding of ionic electrolyte in energy storage field. It could be accepted by Materials after addressing the following comments:
- For the Figure 1, although EDLC, redox pseudocapacitance, and the hybrid ones are mentioned, it may not be comprehensive enough, recently there are some intercalation pseudocapacitance proposed and validated, mainly the cations intercalated into layered metal oxide such as Nb2O5 with strong redox while almost no lattice structure variation, so it's better to include this as well.
- After the EES devices, the authors covers the aqueous, organic and ionic electrolytes, which are liquid based electrolytes. It could be better to have a short paragraph talking about solid electrolyte, polymer electrolyte and liquid electrolyte to compare these electrolyte properties at different physical states, and then afterwards, moving to the section of aqueous, organic and ionic electrolytes.
- Since all different types of cations in ILs are listed, it is better to have a paragraph to describe Pyridinium and Phosphonium based ILs with a few examples.
Author Response
Please find the respond to the comments of the Reviewer 3 in the attached file.

Round 2
Reviewer 1 Report
The authors properly responded my comments. The manuscript can be accepted in the current format.
Reviewer 3 Report
My comments are well addressed, it is ok for me to publish.